# Policy Implications for Protecting Health from the Hazards of Fire Smoke. A Panel Discussion Report from the Workshop *Landscape Fire Smoke: Protecting Health in an Era of Escalating Fire Risk*

**DOI:** 10.3390/ijerph18115702

**Published:** 2021-05-26

**Authors:** Christine T. Cowie, Amanda J. Wheeler, Joy S. Tripovich, Ana Porta-Cubas, Martine Dennekamp, Sotiris Vardoulakis, Michele Goldman, Melissa Sweet, Penny Howard, Fay Johnston

**Affiliations:** 1Centre for Air Pollution, Energy and Health Research (CAR), Liverpool 2170, Australia; amanda.wheeler@acu.edu.au (A.J.W.); joy.tripovich@sydney.edu.au (J.S.T.); ana.porta@sydney.edu.au (A.P.-C.); Martine.Dennekamp@epa.vic.gov.au (M.D.); fay.johnston@utas.edu.au (F.J.); 2South West Sydney Clinical School, The University of New South Wales, Liverpool 2170, Australia; 3Ingham Institute of Applied Medical Research, Liverpool 2170, Australia; 4Woolcock Institute of Medical Research, The University of Sydney, Glebe 2031, Australia; 5Mary MacKillop Institute for Health Research, Australian Catholic University, Melbourne 3000, Australia; 6Environmental Public Health, Environment Protection Authority Victoria, Melbourne 3004, Australia; 7National Centre for Epidemiology and Population Health, Australian National University, Canberra 2601, Australia; Sotiris.Vardoulakis@anu.edu.au; 8Asthma Australia, Chatswood 2067, Australia; MGoldman@asthma.org.au; 9Croakey Health Media, Sydney School of Public Health, The University of Sydney, Sydney 2006, Australia; melissa@croakey.org; 10Maritime Union of Australia, West Melbourne 3003, Australia; penny.howard@mua.org.au; 11Anthropology Department, The University of Sydney, Sydney 2006, Australia; 12Menzies Institute for Medical Research, University of Tasmania, Hobart 7000, Australia

**Keywords:** air pollution, PM, landscape fires, bushfires, smoke, health risk, exposure, communication, protection, policy

## Abstract

Globally, and nationally in Australia, bushfires are expected to increase in frequency and intensity due to climate change. To date, protection of human health from fire smoke has largely relied on individual-level actions. Recent bushfires experienced during the Australian summer of 2019–2020 occurred over a prolonged period and encompassed far larger geographical areas than previously experienced, resulting in extreme levels of smoke for extended periods of time. This particular bushfire season resulted in highly challenging conditions, where many people were unable to protect themselves from smoke exposures. The Centre for Air pollution, energy and health Research (CAR), an Australian research centre, hosted a two-day symposium, *Landscape Fire Smoke: Protecting health in an era of escalating fire risk*, on 8 and 9 October 2020. One component of the symposium was a dedicated panel discussion where invited experts were asked to examine alternative policy settings for protecting health from fire smoke hazards with specific reference to interventions to minimise exposure, protection of outdoor workers, and current systems for communicating health risk. This paper documents the proceedings of the expert panel and participant discussion held during the workshop.

## 1. Background

Globally, bushfires are expected to increase in frequency and intensity due to warmer and drier conditions because of anthropogenic climate change [1,2,3,4]. This is also expected to be the case for Australia [5]. Indeed, recent research on climate and fire risk modelling has reported that anthropogenic climate change has increased the probability of intense fires in south-east Australia (as seen during the 2019–2020 period) by 30%, and states that this is an underestimate [6].

Particulate matter (PM) is the predominant air pollutant and is often emitted in highly elevated concentrations during bushfire or prescribed burning events (planned burns which are conducted to minimise loss during subsequent bushfire episodes). Historical records of extreme PM pollution in Sydney from 1994 to 2007 reported that for 67% of peak pollution days where a cause was identified, 94% were attributed to bushfires or prescribed burning events [7]. Similar results were found for other Australian cities in subtropical and arid locations [7]. Although bushfire smoke is characterised by high levels of PM, both coarse (PM_10_) and fine PM (PM_2.5_), as well as aldehydes, volatile organic compounds and carbon monoxide [8], health-based advice during bushfire events is predominantly driven by PM concentrations. This is because PM pollution can affect very large areas and be present many kilometres from the fire source, and thus potentially affect large populations, compared to the other volatile or gaseous pollutants which are present close to the fire source.

Fine particles (PM_2.5_) can penetrate deep into the lungs and cause inflammation. They are also able to enter the blood stream to affect different body systems. PM_2.5_ smoke-related emissions are associated with multiple health impacts [1], with no safe exposure thresholds currently established. These include adverse respiratory outcomes, especially in people with Chronic Obstructive Pulmonary Disease (COPD) [9,10,11,12,13] and asthma [10,11,12,13,14,15]; and adverse cardiovascular outcomes including ischemic heart disease, heart failure and cardiac arrest [10,11,15,16,17]. The association between smoke exposure and increased mortality is mixed, although evidence of an effect is mounting from more recent studies [1,18,19], and an Australian study estimated that there were 417 excess deaths in south-east Australia attributable to the 2019–2020 bushfire season [20].

People with existing medical conditions (particularly respiratory and cardiovascular conditions), children, women who are pregnant and older people, are considered to be most at risk of being affected by bushfire smoke [21]. Australia has a high prevalence of people with COPD (~30% of those aged ≥75 years) [22], asthma (11%) [23], and cardiovascular disease (5.6% of those aged >18 years) [24]. These groups, along with our ageing population, mean that many in our community are at higher risk of experiencing adverse effects of bushfire smoke.

### 1.1. Current Health Protection Advice

To date, health protection advice has been issued to the public when unplanned bushfires or prescribed burning activities have occurred. The dissemination of this information has taken place via government websites of mainly health [25,26] and environment agencies [27,28]. General advice has aimed to mitigate health impact by reducing exposures to bushfire smoke by recommending specific actions such as:(a)Closing all doors and windows and airing out the house when air quality improves;(b)Using air conditioning systems on a recirculating mode (if available); (c)Avoiding strenuous activities or exercising outdoors; and(d)Taking shelter in an air-conditioned public building or re-locating to less polluted areas.

Advice has also centred on reducing the potential and severity of impact prior to exposure occurring by advising susceptible population sub-groups to take medications for respiratory or cardiovascular disease if and as prescribed by medical practitioners.

However, there are limitations of public health messaging around bushfire smoke hazards as the advice is focused on mitigation measures for short-term exposures only and relies heavily on individual-level action [29]. Furthermore, there is concern that messaging is not sufficiently targeted to affected geographic areas. This has led some researchers to advocate for alternative strategies for messaging based on location-specific air quality forecasts and studying hourly PM_2.5_ concentrations at nearby air quality monitoring locations, given that PM_2.5_ concentrations can vary widely across large urban areas during peak pollution events [21]. This targeted information could then be communicated to the public to allow planning of daily activities in ways that minimise exposure to pollution [21].

### 1.2. 2019–2020 Bushfires in Australia 

Even for a country thought to be well prepared to mitigate and combat bushfires, the catastrophic 2019–2020 bushfires in Australia were unprecedented in scale. These fires caused 33 direct deaths [30] and are estimated to have been associated with 417 excess deaths, 1305 emergency department admissions for asthma and 3151 hospital admissions due to smoke exposure [20]. Furthermore, there were more than 3000 homes destroyed and an estimated 24–40 million hectares burnt across various states, with 5.6 million hectares burnt in the state of NSW alone [30]. It is also estimated that three billion animals (comprising mammals, birds, reptiles and frogs) either died or were displaced [30,31], and additional extensive damage to property and the environment was also documented [21]. The fires occurred across Australia from August 2019 until February 2020 and impacted both the east and west coasts [30]. In addition to these direct adverse health outcomes, the fires resulted in ongoing exposure to prolonged elevated particle pollution levels for populations living in our largest cities, including Sydney (approximately 5.34 million), Melbourne (5.16 million) and Canberra (431,000) [32], as well as high exposures for people living in regional and rural areas. A study of the health burden associated with smoke exposure during the 2019–2020 bushfires calculated that PM_2.5_ concentrations (24 h averages) measured across eastern Australia (NSW, Victoria, QLD and the ACT) exceeded the 95th percentile of previously recorded mean concentrations from at least one monitoring station on 125 of 133 (94%) days [20]. This same study calculated that the highest population weighted exposure to PM_2.5_ was 98.5 µg/m^3^, far higher than the national standard of 25 µg/m^3^ and 14 times the historical population weighted mean of 6.8 µg/m^3^ (24 h average) [20]. Although current PM_2.5_ standards cite a 24 h standard of 25 µg/m^3^ as an acute exposure, during the Black Summer bushfires PM_2.5_. 24 h averaged concentrations exceeded 100 µg/m^3^ on many days at monitoring sites in Sydney and regional sites in NSW [33]. In addition, while not directly comparable to the 24 h PM_2.5_ standard, hourly PM_2.5_ concentrations reached up to 800 µg/m^3^ in Sydney and over 2000 µg/m^3^ in regional NSW [33]. PM_2.5_ concentrations in Canberra and Victoria were also highly elevated on occasions during January 2020 [33]. Prolonged and elevated pollution smoke events affecting such large populations are unprecedented in Australia [5,30]. 

Given the extent and magnitude of the 2019–2020 bushfires, the Commonwealth of Australia held an Inquiry to document the “lessons to be learned in relation to the preparation and planning for, response to and recovery efforts“ following the bushfire season [30]. An interim report recommended that increased and ongoing funding be provided to the Department of Health to fund research into the health impacts of bushfire smoke with specific funding allocated to studying the effects on pregnant women, the foetus and infants (Recommendation 6) [30] (https://parlinfo.aph.gov.au/parlInfo/download/committees/reportsen/024518/toc_pdf/LessonstobelearnedinrelationtotheAustralianbushfireseason2019–2020.pdf;fileType=application%2Fpdf (accessed on 15 February 2021)). The Royal Commission into National Natural Disaster Arrangements also published a report in late 2020 [5]. It included a comprehensive chapter on air quality, noting the lack of real time reporting of data and the variability of reporting across jurisdictions, and incorporated a recommendation urging national consistency in how air quality is reported (Recommendation 14).

The NSW Government commissioned an independent inquiry into the 2019–2020 bushfire season. It made similar recommendations to the Commonwealth Government. It recommended that the government invest in health research on the long-term effects of exposure to bushfire smoke, especially for vulnerable groups, that it increase its capacity for air quality forecasting and issuing of alerts (including alerts for other air quality incidents), and that it develop a public education campaign by improving messaging (recommendations 34 and 35) [34]. Inquiries and or reviews into the 2019–2020 bushfires were also held in the states of Victoria [35] (https://www.igem.vic.gov.au/fire-season-inquiry/inquiry-reports/inquiry-phase-1 (accessed on 2 March 2021)). Queensland [36] (https://www.igem.qld.gov.au/queensland-bushfires-review-2019-20 (accessed on 2 March 2021)) and South Australia [37] (https://www.safecom.sa.gov.au/independent-review-sa-201920-bushfires/ (accessed on 2 March 2021)).

Researchers from the Centre for Air pollution, energy and health Research (CAR), a nationally funded research centre, have been actively involved in studying the effects of bushfire smoke exposure over the last two decades [9,10,11,13,14,16,17,20,38,39,40,41,42,43,44]. During and after the 2019–2020 bushfire season, CAR released position papers [45,46], made submissions to the government inquiries [47,48,49] and released a factsheet [50]. Previous research by CAR recognises the short-term health impacts of bushfire smoke as wide ranging, but predominantly manifested as increased respiratory morbidity and mortality outcomes. However, the 2019–2020 bushfire season highlighted the lack of knowledge over the likely longer-term impacts of exposure to bushfire smoke given the scarcity of evidence. From an upstream perspective, CAR’s general position is that a key strategy for reducing community exposure to bushfire smoke is to reduce the severity and frequency of bushfires, and to that end, tackling climate change is essential. On a more local level, other strategies recommended by CAR at the time and in response to the bushfire inquiries included altering building standards and infrastructure to prevent smoke infiltrating the indoor environment; improving public communication and alert systems around air quality; and providing nationally consistent air quality reporting to improve public awareness with an emphasis on PM_2.5_ concentrations instead of the Air Quality Index (AQI) (an index which amalgamates data on several air pollutants) [46]. Given CAR’s acknowledgement of the difficulties associated with reporting and communicating health-related information, especially in a fast-paced, dynamic and changing environment, CAR was interested in broadening the discussion around bushfire smoke exposure mitigation and public education needs. 

As part of its knowledge brokering and translation activities and prior to the 2019–2020 bushfires occurring, CAR had scheduled a workshop for early 2020, to bring together experts and interested parties, to present on and to discuss the science and policy around the health impact of bushfire smoke exposure. The bushfire events of the 2019–2020 summer season and the ensuing Inquiries only served to highlight the relevancy of such a workshop. The aim of this paper is to document the proceedings of the expert panel and participant discussion held during the workshop. 

## 2. Workshop Panel Discussion 

The Centre for Air pollution, energy and health Research (CAR) held a workshop, *Landscape Fire Smoke: Protecting health in an era of escalating fire risk*, on 8 and 9 October 2020. The workshop had been scheduled by CAR as part of its knowledge brokering and translation activities. While the catastrophic nature of the 2019–2020 summer bushfire season highlighted the pertinency of the workshop, the second crisis in the form of the COVID-19 pandemic followed swiftly afterwards, and given the widespread disruption to work patterns and the inability to hold large meetings face to face, the workshop was moved to an online format using Zoom as a delivery platform. 

Given that there are many factors impacting on public health response during major bushfire events, including existing policies and practices and the varying responsibilities of response agencies (both front line emergency responders and others), the workshop aimed to engage and invite attendees from a wide range of disciplines. The aim of the workshop was to share information and research findings on the health impacts of exposure to bushfire smoke and to engage in discussion and debate over pertinent issues related to the overall public health response to the bushfire crisis. 

Two methods/activities were used to help facilitate discussion and solicit opinions during the workshop: (1) engagement of experts for a panel discussion; and (2) the use of “polling” questions to the wider audience of workshop participants. The panel discussion is documented in this paper to help inform future research and policy in this domain. The polling questions are presented in the Appendix A. Further detail is provided below. 

### The Expert Panel 

Given the concern over the adequacy of existing policy and practice in public health response to bushfires, CAR sought to incorporate free form but guided panel discussion asking an expert panel to respond to questions. The expert panel consisted of four experts representing diverse professional domains, all with an interest in bushfire health impacts, and/or who represented groups which had been impacted by the 2019–2020 bushfire crisis. Briefly, they represented: the CEO of Asthma Australia, a national advocacy group for people with asthma and respiratory health issues (Michele Goldman); an environmental health professor and researcher from the Australian National University (ANU) (Professor Sotiris Vardoulakis); a representative from the Maritime Workers Union (MUA), a union representing a large outdoor occupational group (Dr Penny Howard); and a communications expert and Managing Editor of Croakey Health Media, an online public health journalism forum, and honorary academic at the School of Public Health, The University of Sydney (Dr Melissa Sweet). The intent of the expert panel was to draw a range of viewpoints from informed experts who could highlight difficulties faced by the public during the bushfire crisis, including potentially susceptible sub-groups, and to discuss issues associated with occupational exposure. Further detail on each of the expert panel members is available in the Appendix A.

The purpose of the panel discussion was to discuss the adequacy of current policies and practice in public health response to bushfires and prescribed burn events. A moderator, Associate Professor Fay Johnston, a Chief Investigator with CAR, was appointed to ask the questions and moderate the discussion. Dr Johnston has a long-standing research program studying the effects of bushfire and wood smoke effects on health. Specifically, the panel and audience were asked:
(i)How useful are the current recommended standard interventions for protecting human health and what other approaches should we consider?(ii)How do we protect outdoor workers?(iii)Do our current systems work for communicating air quality and health risk?

Two scientists affiliated with the Centre for Air pollution, energy and health Research (CAR) took notes during the panel and audience discussion to fully capture all responses. Attendees were invited to post questions in the “Chat” function of the Zoom software being used, and panel members were invited to respond.

In addition to the expert panel questions, a series of “polling” questions were asked of online participants in the form of multi-answer responses. Workshop participants represented wide-ranging backgrounds from academia, research, government, and advocacy groups. The main aim of the polling questions was to solicit responses from the workshop participants to determine whether there was commonality in the concerns, opinions and responses given. The responses to the polling questions were immediately tallied and the percentage response to each category were displayed online as feedback to all participants. A secondary, but equally important aim of the polling questions was to trigger further discussion from the expert panel. The polling questions and responses are provided in the Appendix A.

## 3. Panel Discussion Findings

The workshop attracted 86 attendees including the expert panel members. Table 1 groups the agencies that participants were drawn from and the number of participants for each category. By far the greatest proportion of participants comprised researchers and academics (70%), followed by government employees (19%) and personnel from advocacy groups (7%).

This paper presents the findings of the discussion in three main themes which align with the questions discussed: (1) efficacy of current interventions for protection of health; (2) protection of workers; and (3) communication of air quality and health risk during smoke episodes. Although there was sometimes overlap of discussion across themes, we have grouped similar discussions for coherency. 

### 3.1. Theme 1. Efficacy of Current Interventions for Protection of Health

In general, current public health advice during bushfire events requires individual-level actions to protect health. These include, but are not limited to, avoiding strenuous exercise, staying indoors, and using air conditioners to recirculate indoor air. This advice has a limited evidence base for reducing exposure and is based on virtually no evidence regarding health protection [29,51]. We therefore asked the expert panel to respond to the questions below and workshop attendees were asked to respond to specific polling questions online. 

**Question** **1.**
**How useful are the current recommended standard interventions for protecting human health and what other approaches should we consider?**


The expert panelists noted the importance of health factsheets and information targeting protection of potentially susceptible groups such as the elderly, children and those with pre-existing respiratory and cardiovascular disease. However, the panel members were united in their concern over the communication of and public understanding of air pollution data.

Panel members were unanimous in their concern that much of the advice provided by government during the 2019–2020 bushfires was impractical as it was tailored in response to shorter smoke episodes that last from a few hours to a few days (that is, advice to stay indoors, close windows, etc.). The panel felt that such actions were impractical for longer smoke episodes, such as those experienced during the 2019–2020 bushfires. 

In particular, it was noted that there was no existing public information on the efficacy of wearing masks during such a prolonged bushfire event. This led to media scrutiny and debate on whether face masks could be useful in mitigating exposure, and if so, what type of mask and how it should be worn. Confounding this issue was the scarcity of empirical evidence to support the wearing of face masks in this situation. It was noted that during the 2019–2020 bushfire season, additional health advice was produced which attempted to address the issue of efficacy of face masks. Examples of non-government agencies issuing health advice included the Research School of Population Health at the Australian National University (ANU) which consisted of fact sheets and infographics translated into twelve languages (https://rsph.anu.edu.au/phxchange/communicating-science/how-protect-yourself-and-others-bushfire-smoke (accessed on 5 March 2021)) [52] and a fact sheet produced by CAR (https://www.car-cre.org.au/factsheets (accessed on 5 March 2021)) [50]. 

The panel also noted that given the extent and magnitude of the 2019–2020 bushfires, Asthma Australia conducted a web survey to gather information from people with asthma and the general community on personal experiences arising from bushfire smoke exposure [53]. The survey collected information over six weeks from over 12,000 people and included both qualitative and quantitative questions on symptom reporting, hospital emergency department attendance, absenteeism form work or school, activity participation, anxiety, and what preventive actions people took to minimise exposure. The survey found that respondents cited lack of adequate information including public health messaging, as well as financial constraints, as the major factors which impacted on their ability to reduce smoke exposure. Some of the recommendations arising from the survey results included provision of better public education around air pollution, better and more consistent air quality reporting, specific advice on the efficacy of face masks, and consideration of how modifications might be made to houses and public buildings. 

While a main health intervention during bushfire episodes is to stay indoors, panel members noted that the type and quality of the building used to house or shelter people is important. The panel noted for instance, that older buildings are prone to greater ingress of outdoor air to the indoor environment as they are “leakier”. Newer buildings provide increased protection as they are better sealed, although, newer research shows that poorly performing houses, with respect to energy efficiency and ventilation, are still being constructed [54]. The panel also raised the issue of the build-up of indoor air pollutants during smoke events, such as carbon dioxide (CO_2_), and the need to at some stage ventilate the building to release the indoor pollutants, if the doors and windows have been kept closed. However, the challenge is in knowing when to open the building for ventilation. Having access to real-time air quality data was suggested as a method to help people decide on such actions, which highlighted the importance of having access to this type of information. Furthermore, there is still limited evidence on the efficacy of air conditioning systems.

Panel members commented on the utility of air cleaners and suggested that more research is needed to demonstrate the most effective interventions. The panel acknowledged that air cleaners could potentially be a valuable tool for health protection during fires, although noted that further research is required to test their efficacy during smoke events. Further, the panel noted that there are several caveats to recommending wide-scale use of air cleaners, including cost, understanding the size of the space to be filtered and managed, and the need to regularly replace filters so that the units function optimally. Panel members supported the use of air cleaners with high-efficiency particulate air (HEPA) filters in work environments and schools and suggested that they could be funded in public schools. While evidence points to a cost–benefit ratio which supports the use of air cleaners as a successful intervention to reduce particle pollution exposure, the panel acknowledged that subsidisation of air cleaners for low-income families might be difficult to implement as a protective measure. 

The panel noted that portable air quality monitoring devices and sensors have been increasingly adopted in a variety of settings and might be valuable in responding to bushfire smoke episodes as well. These devices provide the opportunity to obtain more personalised air pollution exposure data, which panel members agreed was useful. Portable “low-cost” air quality monitoring devices and sensors generally include a proprietary or custom-built particle sensor packaged alongside with other sensors or probes which measure, for example, temperature, humidity, carbon dioxide, or carbon monoxide [55,56]. Advantages of these devices can include their cost, portability, relative ease of deployment and accessibility. The panel acknowledged that for optimal sensor use it is important to educate the community on how to use the data recorded, and the limitations of the data collected. For example, sensor data are useful in providing information on relative changes in air pollution concentrations, rather than absolute changes in concentrations, and sensors require pre and post calibration before use. The panel noted that several recent applications of sensor/device networks in NSW have included citizen science projects involving: children at schools collecting and using the data as a learning tool [57]; specific data gathering in community settings; and collection of data by local councils [58]. 

Another application of sensor technology includes the use of air pollution sensors incorporated into air cleaners, although this currently occurs only in a small minority of cleaners. These sensors can help consumers understand to what extent their air cleaner can decrease or remove particle pollution and also help to educate the user on indoor air quality and environmental conditions. 

### 3.2. Theme 2. Protection of Workers in Occupational Settings

**Question** **2.**
**How do we protect outdoor workers?**


There are substantial gaps in our knowledge on how best to protect outdoor workers during hazardous ambient air pollution events. The panel member representing the Maritime Union of Australia indicated that it has historically had an interest in air quality because ports and their outdoor workplace settings are subject to diesel emission exposures. However, it was acknowledged that the Union was not properly and seriously engaged with the issue of ambient air pollution exposure until faced with the 2019–2020 bushfire season. 

The panel discussed the challenges of having a 24 h workplace, such as exists at ports, and noted that workers are exposed to outdoor air over a full eight-hour or twelve-hour shift. The additional challenge facing port workers is that ships are sometimes docked for only a few hours, and so the turn-around time for unloading and loading goods is sometimes short. 

At the commencement of the 2019–2020 bushfire season, the MUA found that there was a lack of clear health advice relevant to working outdoors in a high pollution setting and within a high-pressure environment. At that time, the MUA obtained advice from various health and environmental researchers on air quality indices (AQIs) and how they relate to PM_2.5_ levels and health effects. However, in the context of the prolonged haze, the Union sometimes struggled to determine how safe the air was for workers. During the bushfire crisis, the MUA chose a level of 37.5 µg/m^3^ PM_2.5_ as a threshold for advising workers to avoid strenuous outdoor work, as this approximated the NSW AQI level of 150, which advised people to ‘avoid strenuous outdoor activity’. During the 2019–2020 bushfire season the MUA used the AirRater app (www.AirRater.org (accessed on 5 March 2021)) to access localised air quality information to use as triggers for action. AQIs were used by government agencies to help characterise health risk associated with the total air pollution mixture present, combining data on the major air pollutants (PM, nitrogen dioxide, ozone) to calculate an index. Two complications were that there was inconsistency in how the various states reported their AQIs [5,21,46] and the evidence on health effects from PM_2.5_ exposure could not be directly related to the AQIs [46]. Since the release of the various inquiry reports, the Australian Health Protection Principal Committee has endorsed nationally consistent PM_2.5_ hourly air quality categories and public messaging (to replace the use of AQIs), and 24 h categories and forecast messaging, with the aim for implementation by all jurisdictions before the 2021–2022 bushfire season [59].

The MUA also developed a draft hierarchy of controls on how risk should be managed. This was communicated to employers and implemented by workplace Health and Safety Representatives (HSRs) a worker elected and then trained under the Workplace Health & Safety Act). Workplace HSRs took the advice to workplace safety committees, and also constantly monitored air quality using AirRater and informed their fellow workers and employers when the threshold was reached. HSRs met in February 2020 to review the union advice and noted that workers had started to experience adverse effects at a PM_2.5_ concentration of 25 µg/m^3^, and agreed that all outdoor work should cease at 37.5 µg/m^3^ PM_2.5_ (there was previously ambiguity for some groups of workers). For instance, it found that elevated concentrations over shorter time frames could occasionally be accommodated by shifting some of the work to alternative times, although such practices required flexibility and clear guidance. The hierarchy of controls outlined in the 2019 advice to members included guidance to elevate action such as talk to your delegate; move sensitive groups indoors; stop more strenuous work (lashing containers lifting and securing very long heavy bars); then stop other work.

However, the MUA found that while this framework was theoretically sound, when applied in the high-pressure work environment, the language needed to be strengthened, well organised, specific and clear to ensure that it was well understood and adhered to. For instance, it found that previous messaging lacked the gravitas and impetus needed to adequately protect workers; using ‘softer’ wording such as “consider minimising” was not direct enough to get organisations/companies to respond. The panel highlighted the need for very clear, unambiguous and consistent wording across jurisdictions to enable organisations to take appropriate actions to protect workers.

During the 2019–2020 bushfire smoke season, the MUA also found that indoor spaces and hence indoor workers were also impacted from the bushfire smoke because of the extreme smoke haze, and so recommended that indoor workplace settings need to be considered in any guidance document.

The panel discussed the practicalities and utility of workers wearing monitors in the workplace. The workshop heard that the MUA is considering installing air quality monitors at the larger port terminals. However, the key issue for the industry is not necessarily in accurately monitoring air quality, but rather it needs to have consensus and agreement on which concentrations are safe or unsafe, and over what exposure period. 

Panel members noted that the difficulties faced by the MUA provided a clear example of how occupational environments are not specifically considered in current public health protection advice provided by the government. Some panel members also identified that there is a need to address vulnerable groups of people working outdoors in addition to outdoor workers in general. However, the MUA noted the importance of considering sensitivities and cultures within this occupational sector as the Union found that victimisation of individuals could occur for workers who chose to remove themselves from the risk. The MUA panel member noted that a key strategy is to have a pre-determined consensus on what is considered safe and to enable a framework which allows workers to take protective actions. The panel agreed that industry and political consensus is needed across all workplaces as it is not sustainable for one industry to act alone on this issue, and ultimately, all employers are required to provide a safe workplace environment under the WHS Act.

### 3.3. Theme 3. Communication of Air Quality and Health Risk

**Question** **3.**
**Do our current systems work for communicating air quality and health risk?**


#### 3.3.1. How AQ Standards Are Implemented and Reported

There was support from the panel for reporting hourly air pollutant concentrations because of temporal fluctuations and the need to plan activities to minimise exposure. However, the panel noted that during a major smoke event, people will be exposed on a continuous basis over hours, days or even weeks, as occurred during the 2019–2020 bushfire season. Therefore, the panel felt it was important to consider how we apply air quality standards and other guidelines. Australia’s air quality standards are implemented by each state jurisdiction and reference World Health Organisation guidelines, and at present do not include an hourly value for particulate matter (24 h and annual average guidelines for PM (PM_2.5_: 24 h average 25 µg/m^3^, annual average 8 µg/m^3^; PM_10_: 24 h average 50 µg/m^3^, annual average 25 µg/m^3^) [60]. After the 2019–2020 bushfire season, some states, such as NSW, have moved to adopt hourly reporting of PM_2.5_ in response to the various inquiry recommendations and community concern.

As hourly concentrations reflect acute exposure, panel members expressed a need to have access to consistently reported hourly averaged air quality data. However, the panel also noted that it is important that the community understands that hourly averaged data will be expressed as a higher value than a 24 h average or annual average concentration. Therefore, greater awareness is needed on how acute exposure guidelines are applied, versus short-term guidelines (24 h averages) and long-term guidelines (annual averages). The panel was of the view that having different exposure guidelines, including an acute (hourly average) guideline, are useful tools to help people manage extreme and fluctuating exposures. The panel provided an example of where hourly concentrations provide only one piece of the exposure puzzle. Woodsmoke can exert both acute and seasonal exposures, particularly in cold-climate settings in Australia. 

From an occupational perspective, Safe Work Australia and SafeWork NSW, government agencies with responsibility for workplace safety, have issued guidance on safe ambient air pollution exposure, which refers to the air pollution standards. However, the panel expressed concern that the guidance is not sufficiently directive to enable clear mitigative actions during heightened air pollution events. Further discussion on this issue is presented in the previous section under “*Theme 2. Protection of workers in occupational settings”.* In summary, members expressed the need for: employers to reflect on the way that standards are used in the workplace; to consider the wording used in guidance documents; develop pre-determined criteria to guide decision making during events; and to support national consistency in workplace response. 

#### 3.3.2. Messaging for Communities, Population Sub-Groups, and Life Stages

Panel members expressed a view that there are structural issues in how public health matters are addressed and communicated to the public. Panel members acknowledged that there is no silver bullet to satisfactorily communicating about health risks of bushfire smoke exposures, and it is clear that a range of communication tools are needed. Several points arose from this discussion. The first and most obvious is that public health risk is highly complex to communicate to individuals, as there are so many diverse settings, issues, underlying health concerns, biases and audiences at play. Secondly, is the issue of whether we generalise public health advice so that it is relevant for all individuals in the community or whether we target messaging to specific sub-groups.

The extent to which public information needs to be tailored to specific groups to enable both accessibility and thorough understanding needs further consideration. It is particularly important for potentially vulnerable groups. Vulnerability can be conferred because of existing health status, because of genetic susceptibility, because of the heightened nature of exposure or because of factors related to life stage. For instance, vulnerable population groups might be impacted by pollutants which are present at just above average or near background concentrations. Conversely, other population groups might be tolerant of longer-term elevated pollution levels with little discernible adverse impact. How we best respond to this spectrum of health impact is unclear.

In some occupational settings, focussing on only vulnerable groups can sometimes be counter-productive, as seen with the example of the MUA workers mentioned above where vulnerable workers were discriminated against for taking protective action. 

With respect to vulnerabilities due to specific life stage, the panel noted that the general health advice issued during the 2019–2020 bushfire smoke episodes was not sufficient for pregnant women, the very young and the elderly. In particular, it noted that being pregnant and/or caring for young children can be emotionally charged when dealing with and assessing risks. Overall, the panel acknowledged the challenge in providing information which is relevant for the diversity of population sub-groups, and therefore recommended some targeted advice.

An important communication issue arising from the 2019–2020 bushfire season was the need to address concerns over different exposure timeframes, that is, recommended actions for the short, medium and long term. This stemmed from the unprecedented length of time that bushfire smoke impacted various communities and the extent of geographic area impacted. It was noted that the survey conducted by Asthma Australia found that many people were anxious, scared, concerned, and expressed feelings of helplessness that the smoke event went on for too long to be able to implement the advice provided [53]. For instance, it was too difficult to keep doors and windows closed for so many consecutive days/weeks without polluted air ingressing into dwellings. There was also a lack of empirical data on the effects of medium-term exposure (weeks to a couple of months) to elevated bushfire smoke PM, due to the dearth of studies on medium-term exposures. This is contrasted by existing evidence of adverse effects from short-term acute exposures and long-term chronic exposures. These limitations made it difficult to construct public health messages directly relevant to what people were experiencing.

On a contrasting note, appropriate messaging is also needed for healthy members of the population to avoid anxiety, associated with receiving constant advice and alerts, and the acknowledgment of the link between anxiety and depression and poorer health. The panel emphasised the need for public messaging which avoids panic, but which clearly reiterates actions needed to protect personal health. During the bushfires, there was much comparison by the media of the health impact from cigarette smoking compared with exposure to bushfire smoke. This was not found to be particularly helpful as bushfire smoke exposure is unavoidable for most people. It was also not considered to be a valid comparison from a scientific perspective because of the different elemental composition of the particulate pollution deriving from the two distinctive sources.

On the issue of messaging for healthy people, panel members commented that a wholistic approach should be taken which considers a cradle to grave approach. This approach recognises that individuals move through life and experience developmental and physiological stages at various times. This risk is likely to change over time. For instance, even for healthy humans, over time, chronic exposure to lower-level (but still harmful) pollutant concentrations may lead to greater susceptibility compared to episodic exposure to elevated pollutant episodes. Hence, it is important to provide messaging which enables healthy people to understand the potential for health risks from poor air quality at specific points in time, so that they can take preventive action to safeguard health.

The panel highlighted the need for better education of and engagement with the community to help improve environmental health literacy. One mechanism could be a high-profile education campaign. For instance, the panel noted that Asthma Australia considers implementation of an AirSmart education campaign that describes the different air pollutants, their sources and how they impact health, to be the most important strategy in improving community understanding of the effects of air pollution (https://asthma.org.au/about-us/media/new-national-health-campaign-on-air-genda (accessed on 5 March 2021)). Such a strategy could be modelled on the SunSmart campaign which successfully raised awareness of the dangers of risky sun exposure. 

Overall, the panel agreed that there is a need for a systematic communication response around bushfire smoke exposure which helps to provide people with the necessary knowledge around health protection, including awareness of risks and actions that can be implemented over varying timeframes and life stages. 

#### 3.3.3. How Information Is Disseminated

Panel members noted that from a media perspective, much of the current public communication is based on crisis communication. Additionally, sectors of the media and the public continually face a deluge of misinformation and dis-information around risk which can erode public health discourse and messaging. It was noted that sound knowledge on air pollution health effects is often lacking within the health sector, with the risks often not fully appreciated by health care workers and some public health professionals. 

The panel highlighted the power of stories and the need for local journalism to reflect and tell the stories. However, there was concern that local news outlet services have been declining, which have resulted in an information vacuum for some local communities [61]. Social media platforms such as Facebook and Twitter have a role in providing news, but they are insufficient, and there is real concern over the quality of advice and misinformation provided. 

### 3.4. Translating Scientific Evidence into Policy and Practice

We discussed how to make evidence matter and how to entrench evidence into politics and policy. Panel members noted that in Australia, political debate around COVID-19 has been broadly accepting of scientific evidence, and protection of public health has been central to decision making around risks. This, however, has not been the case on issues related to climate change where the debate has been substantially more polarised and where scientific evidence has not been considered with the same gravitas. Panel members suggested that there needs to be reframing of our public debate, primarily to de-politicise the dialogue, and to avoid ideologies from overshadowing scientific evidence. It was suggested that there needs to be transformation of the way we report to and educate the public on air quality as well as other environmental issues. 

The panel also recommended strengthening the evidence base and having better translation of evidence to help influence policy decisions. Evidence can take two forms: one as rigorous scientific studies which are published after peer review; another which incorporates citizen science methods. The panel highlighted that one theme that emerged strongly during the workshop was the need for collective action, and it advocated for the role of citizen science and training of people to create impetus and momentum. Such actions result in a stronger evidence base and enhanced public understanding to press for political change. 

Alternatively, we can also build on the data which are made available to the public. Examples include smartphone apps, such as the AirRater app (https://airrater.org/ (accessed on 5 March 2021)), which is particularly useful in allowing individuals to access simple air quality information and which also allows recording of symptoms over time [62]. This enables individuals to understand how they are impacted by their environment and can lead to a more wholistic approach to managing individual health [63,64]. 

The expert panel argued the need for effective leadership and for governments to adopt a systems approach to protecting public health during bushfire crises. It noted that this had not occurred during the 2019–2020 bushfire season, and felt that instead, there was an emphasis on individual response to manage health concerns. The panel expressed the opinion that individuals should not be left to manage their exposure and health without support. 

Panel members also noted the underlying inequity in society which disadvantages the capacity of lower-income groups and some workers to respond to public health advice and to protect themselves, especially in the case of environmental exposures such as air pollution, climate change impacts and other environmental impacts. There is also value in engaging with Indigenous people, where health is not seen as an individual entity, but instead, a healthy person is considered as part of a healthy community and healthy environment. This Indigenous notion of health broadens our focus from an individualistic perspective to a planetary health perspective, which can help inform society’s response to harmful bushfire smoke episodes. For example, it has been recognised that acting upstream on access to healthy housing for all sectors of society is likely to have benefits across multiple domains. Panel members were of the opinion that to the greatest extent possible, institutional actions taken should be multi-faceted, with systemic solutions that serve the majority of the population.

Finally, a precautionary response was recommended, noting the substantial scientific literature demonstrating the link between smoke exposure and health impacts. The panel advocated for government to work with communities to safeguard health, and expressed the need to invest in community settings such as public schools, child-care centres and aged-care facilities to ensure these spaces are safe for those vulnerable sub-populations. Panel members argued the need to shift social and political priorities to prioritise health outcomes, so that the onus is on a precautionary approach, in light of the existing evidence.

## 4. Reflections on Panel Discussion Methods 

We note that the format of the panel discussion held during the *Landscape Fire Smoke: Protecting health in an era of escalating fire risk* workshop was successful in drawing opinions from four experts representing academia and research, advocacy, an employer’s union/workplace interests and a public health journalist. The relatively short time allocated to the panel discussion solicited robust and frank discussion, and considering the range of expertise and backgrounds represented by the expert panel members, there was substantial concordance in opinions and views. The question prompts were a useful method to help guide discussion, especially given the online format. Furthermore, the “chat” function on the online platform allowed workshop participants to offer advice, their own perspectives, or ask further questions. The online polling questions provided an additional method of soliciting views from the participants. However, the results should be considered in light of the occupations and backgrounds from which participants were drawn.

## 5. Conclusions

The panel discussion indicated that health communication remains a major challenge during bushfire events with much of the conversation centred around how a successful public health response requires a clear and consistent approach to messaging. Specifically, panel members thought that public health advice needed to: include reference to health risk relevant to the timeframe of exposure (i.e., acute, medium term and chronic); be nuanced for and target vulnerable/different sub-groups of the population; and highlight that variability in health response can occur at different life stages. The panel also highlighted the need for advice on protective actions when high exposures occur over extended periods of time in the order of days, weeks, and months, and felt that as a high-risk community in Australia, we are currently insufficiently prepared to deal with such events. Reporting of air quality by government agencies and dissemination for public use was also highlighted as needing improvement, particularly with respect to providing hourly pollutant data and standardising how air quality is reported across jurisdictions. This issue was identified by the various inquiries which also recommended consistent reporting and categorisation of smoke-related air pollution across states in Australia. Panel members were of the strong opinion that better science literacy around health and air pollution is key. These issues were also considered highly relevant for occupational settings, with panel members highlighting the need for national consistency in advice given and a pre-determined framework for decision making on workplace safety during bushfire episodes. 

The panel noted the need for further research on the efficacy of interventions such as using air cleaners and making buildings more air tight, as well as information tools such as using air sensors to provide personal or local data on exposures. 

Finally, panel members favoured a preventive approach to protecting health during bushfire smoke episodes given the sufficient scientific evidence of adverse health impacts of such events. However, this needs to be supported by strong leadership in government which recognises and acts on the upstream impact of climate change on bushfire risk, as well as its impact on vulnerable sub-groups in the community.

## Figures and Tables

**Table 1 ijerph-18-05702-t001:** Number of participants by category.

Audience	Number of Attendees by Category *n* (%)
Advocacy	6 (7%)
Government	16 (19%)
Media	1 (1%)
Politician	1 (1%)
Researchers/Academic	60 (70%)
Technical	1 (1%)
Union	1 (1%)
**Total participants**	**86**

## Data Availability

Not applicable.

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
