# Peer review of "Policy Implications for Protecting Health from the Hazards of Fire Smoke. A Panel Discussion Report from the Workshop Landscape Fire Smoke: Protecting Health in an Era of Escalating Fire Risk"

_ijerph, 2021, doi:10.3390/ijerph18115702_

Round 1

Reviewer 1 Report

The manuscript (MS) deals with an ever more important issue, i.e., protection of human health from wildfire smoke over extended periods (weeks to months), based on the experiences from the Australian summer of 2019-20. It discusses findings from the Centre for Air pollution, energy and health Research (CAR), who hosted a two-day symposium "Landscape Fire Smoke: Protecting health in an era of escalating fire risk", held 8-9 October 2020. The symposium included a dedicated panel discussion where invited experts were asked related questions.

The MS is well written and contains several clues as to how the authorities can improve many aspects, including taking a scientific approach similar to what the AUS authorities (among few Western type countries…) did regarding the COVID-19 pandemic.

Line 1: The MS is presented as an Article. The Abstract does, however, not have the style of a regular scientific article abstract. In particular, the Abstract lacks the main findings (which would be difficult in an abstract totaling 250-300 words). I suggest that you chose, e.g., Case Report, Perspective or any of the other types allowed by MDPI that may fit for the MS as is.

Line 2: Follow the Template, i.e., capital letter, and maybe consider "Suggestions for Improved Policy Settings …".

Line 41: Introduction/Background should be "1. Introduction/Background". See the Template, and correct also for the other Sections.

Line 4: References should be in brackets, i.e., … climate change [1-4]. This … (Check throughout.)

Line 5: … for Australia (CoA, 2020). Should rather be … for Australia [5]. (?) and correction for the other ref. numbers?

Line 78 and Line 103: Are these maybe the 2nd and 3rd subsection of section 1. Introduction/Background? (And then, section 2. Methods, etc?)

Also check the structure for the other (sub)sections, and follow the Template.

Line 78+: References, Check the Template (including italic, bold phase, etc). Add DOI if available.

In the Results and Discussion section, please present a short discussion on the significance and quality of this particular approach for gathering valuable information about suggestions for improvements. Also, if you have any ideas, please also present what you would have done differently – if anything – based on any retrospective considerations. (This may be valuable for others trying a similar approach for other situations in the future.)

The MS is well written and presents a novel approach and much valuable information. Thanks for an interesting read.

Author Response

Response to reviewers-bushfires symposium manuscript

-Keywords: I added “policy” as a keyword.[1]

-We have now changed the format of the manuscript to a “Conference Report”

Reviewer 1.

-Line 1: The MS is presented as an Article. The Abstract does,however, not have the style of a regular scientific article abstract.In particular, the Abstract lacks the main findings (which wouldbe difficult in an abstract totaling 250-300 words). I suggest that you chose, e.g., Case Report, Perspective or any of the othertypes allowed by MDPI that may fit for the MS as is.

We have changed the title of the manuscript from “Policy settings for protecting health from the hazards of fire smoke“ to

Policy implications for protecting health from the hazards of fire smoke. A panel discussion report from the workshop Landscape Fire Smoke: Protecting health in an era of escalating fire risk” - to better reflect the nature of the manuscript as being imparting communication rather than presenting a research project.

Furthermore in the template we have indicated the article is a “Conference report” paper.

-Line 2: Follow the Template, i.e., capital letter, and maybe consider "Suggestions for Improved Policy Settings …".

The journal template has now been used and we have renamed the manuscript as indicated above.

-Line 41: Background should be "1. Background". See the Template, and correct also for the other Sections.

Thank you, the sections have been numbered as per the template.

Section 2 (line 105). Title of this section has been changed from “Methods” to “Workshop panel discussion” (according to MDPI formatting of “Conference Report” manuscripts).

Wording at the end of this section (lines 227-9) has been changed from/to “The panel discussion is documented in this paper to help inform future research and policy in this domain. forms the basis of the documentation from the workshop which is presented in this paper.

The section titled “Results and Discussion” has been reworded to “Panel Discussion Findings” and has been numbered as section “3.” (line 368).

-Line 4: References should be in brackets, i.e., … climate change[1-4]. This … (Check throughout.)

The references have been amended to be in accordance with the journal template.

-Line 5: … for Australia (CoA, 2020). Should rather be … forAustralia [5]. (?) and correction for the other ref. numbers?

This reference has been corrected and all references checked.

-Line 78 and Line 103: Are these maybe the 2 and 3subsection of section 1. Introduction/Background? (And then,section 2. Methods, etc?)

These are sub-sections of Section 1 Background, and the other sections renamed and numbered as indicated above.

-Also check the structure for the other (sub)sections, and follow the Template.

This has been checked and amended where needed, as indicated above.

-Line 78+: References, Check the Template (including italic, boldphase, etc). Add DOI if available.

The references have been checked and formatted in accordance with the journal requirements.

-In the Results and Discussion section, please present a short discussion on the significance and quality of this particular approach for gathering valuable information about suggestions for improvements. Also, if you have any ideas, please also present what you would have done differently – if anything –based on any retrospective considerations. (This may be valuable for others trying a similar approach for other situations in the future.)

A new section has been created – “4. Reflections on panel discussion methods”. We have moved the first paragraph from the conclusion section to this section and have expanded the discussion to read:

“We found that the format of the panel discussion held during the - Landscape Fire Smoke: Protecting health in an era of escalating fire risk workshop - was successful in drawing opinions from four experts representing academia and research, advocacy, an employer’s union/workplace interests and a public health journalist. The relatively short time allocated to the panel discussion solicited robust and frank discussion, and considering the range of expertise and backgrounds represented by the expert panel members, there was substantial concordance in opinions and views. The question prompts were a useful method to help guide discussion, especially given the online format. Furthermore, the “chat” function on the online platform allowed workshop participants to offer advice, their own perspectives, or ask further questions. The online polling questions provided an additional method of soliciting views from the participants, however, the results should be considered in light of the occupations and backgrounds from which participants were drawn.”

-The MS is well written and presents a novel approach and much valuable information. Thanks for an interesting read.

Thank you.

Reviewer 2 Report

The paper is well written and addresses an important and highly relevant topic. 

My suggestions are very minor, as follows:

Line 54: suggest using subscript formatting of numerals in PM10 and PM2.5 as you have elsewhere.

Line 64: missing space after established

Lines 73-77 This is a long sentence, consider revising.

Lines 84-89:  Suggest providing references for these points.

Line 115: Suggest that you qualify "extremely elevated" or present a range.

Line 191: Please correct formatting of quotation marks.

Line 195: The credentials of the panel members might be better included as supplementary material.

Line 346: I would argue that even new Australian homes are too leaky and we still have a long way to go (as you mention in your point about building standards on line 157). This topic could be extended. Please see: Ambrose, M.; Syme, M. Air tightness of New Australian residential buildings. Procedia Eng. 2017180, 33–40.)  

Author Response

-The paper is well written and addresses an important and highly relevant topic.

Thank you for these comments.

-My suggestions are very minor, as follows:

-Line 54: suggest using subscript formatting of numerals in PM10and PM2.5 as you have elsewhere.

Thank you, we have amended as suggested.

-Line 64: missing space after established

Amended.

-Lines 73-77 This is a long sentence, consider revising.

We have revised this statement and split into two sentences to read (lines 77-82):

“Australia has a high prevalence of people with COPD (~30% of those aged ≥ 75 years) [2], asthma (11%) [3], and cardiovascular disease (5.6% of those aged >18 years) [4]. These groups, along with our ageing population, means that many in our community are at higher risk of experiencing adverse effects of bushfire smoke.”

-Lines 84-89: Suggest providing references for these points.

We have added references as suggested (lines 87-88).

-Line 115: Suggest that you qualify “extremely elevated” or present a range.

We have added changed that wording to “prolonged elevated” and added text as follows (at lines 135-148):

A study of the health burden associated with smoke exposure during the 2019-2020 bushfires, calculated that PM2.5 concentrations (24 hour averages) measured across eastern Australia (NSW, Victoria, QLD and the ACT) exceeded the 95th percentile of previously recorded mean concentrations at at least one monitoring station on 125 of 133 (94%) days. This same study calculated that the highest population weighted exposure to PM2.5 was 98.5 ug/m3, far higher than the national standard of 25 ug/m3 and 14 times the historical population weighted mean of 6.8 ug/m3 (24 hour average). Although current PM2.5 standards cite a 24 hour standard of 25 ug/m3 as an acute exposure, during the Black Summer bushfires PM2.5  24 hour averaged concentrations exceeded 100 ug/m3 on many days at monitoring sites in Sydney and regional sites in NSW. While not comparable to the 24 hour average PM2.5  standard, hourly PM2.5 concentrations reached up to 800 ug/m3 in Sydney and over 2000 ug.m3 in Goulburn. PM2.5 concentrations in Canberra and Victoria were also highly elevated on occasions during January 2020.

-Line 191: Please correct formatting of quotation marks.

This has been corrected.

-Line 195: The credentials of the panel members might be better included as supplementary material.

The majority of the credentials of the Expert Panel have been moved to the supplementary material, and the text has been slightly modified in the manuscript to reflect this as follows (lines 237-248):

 “Briefly, they represented: the CEO of Asthma Australia, a national advocacy group for people with asthma and respiratory health issues (Michele Goldman); an environmental health professor and researcher from the Australian National University (ANU) (Professor Sotiris Vardoulakis); a representative from the Maritime Workers Union (MUA), a union representing a large outdoor occupational group (Dr Penny Howard); and a communications expert and Managing Editor of Croakey Health Media, an online public health journalism forum, and honorary academic at the School of Public Health, The University of Sydney (Dr Melissa Sweet). The intent of the expert panel was to draw a range of viewpoints from informed experts who could highlight difficulties faced by the public during the bushfire crisis, including potentially susceptible sub-groups, and to discuss issues associated with occupational exposure. Further detail on each of the Expert Panel members is available in the Supplementary Information.”

-Line 346: I would argue that even new Australian homes are too leaky and we still have a long way to go (as you mention in your point about building standards on line 157). This topic could beextended. Please see: Ambrose, M.; Syme, M. Air tightness of New Australian residential buildings. Procedia Eng. 2017, 180,33–40.)

We have made reference to this reference and that the study found that some newer Australian houses are still being poorly constructed form an energy and ventilation perspective.  The new text (lines 428-432) reads:

 “While a main health intervention during bushfire episodes is to stay indoors, panel members noted that the type and quality of the building used to house or shelter people is important. The panel noted for instance, that older buildings are prone to greater ingress of outdoor air to the indoor environment as they are “leakier”. Newer buildings provide increased protection as they are better sealed, although, newer research shows that poorly performing houses, with respect to energy efficiency and ventilation, are still being constructed (ref)”.

 The text reflects the panel discussion, so we did not feel it appropriate to expand more on this issue in this manuscript.

Reviewer 3 Report

This is nothing more than workshop report, not for journal.

Author Response

-This is nothing more than workshop report, not for journal.

We have checked with the sub-editor and the journal allows for “Conference Reports”. We have therefore, reframed the manuscript as a Conference Report, as indicated in our response to Reviewer 1.

 Separate comment to the Editor from Reviewer 3:

-They need to summarise more, and have to provide some solution.

The journal does not have a word limit for manuscripts and so we have kept the text as is after responding to the reviewer 1 and 2’s comments.

-Conference report should be providing the findings, rather than simply describing what is discussed.

We feel it is appropriate to describe the workshop discussion to reflect the panel discussion as we have presented it in the manuscript. The purpose of the manuscript is to make public the discussion from the workshop to help inform policymakers and practitioners of expert concerns around protection of health from bushfire smoke exposure. Government looks to published data and information to help inform policy and we feel this manuscript provides solid evidence for further action.

Round 2

Reviewer 3 Report

Looks good now.